# Dual Role of Chondrocytes in Rheumatoid Arthritis: The Chicken and the Egg

**DOI:** 10.3390/ijms21031071

**Published:** 2020-02-06

**Authors:** Chia-Chun Tseng, Yi-Jen Chen, Wei-An Chang, Wen-Chan Tsai, Tsan-Teng Ou, Cheng-Chin Wu, Wan-Yu Sung, Jeng-Hsien Yen, Po-Lin Kuo

**Affiliations:** 1Graduate Institute of Clinical Medicine, College of Medicine, Kaohsiung Medical University, Kaohsiung 80708, Taiwan; 990331kmuh@gmail.com (C.-C.T.); chernkmu@gmail.com (Y.-J.C.); 960215kmuh@gmail.com (W.-A.C.); 2Division of Rheumatology, Department of Internal Medicine, Kaohsiung Medical University Hospital, Kaohsiung 80756, Taiwan; d10153@ms14.hinet.net (W.-C.T.); tsanteng@yahoo.com.tw (T.-T.O.); wucc@cc.kmu.edu.tw (C.-C.W.); hemidark@yahoo.com.tw (W.-Y.S.); 3Department of Physical Medicine and Rehabilitation, Kaohsiung Medical University Hospital, Kaohsiung 80756, Taiwan; 4School of Medicine, College of Medicine, Kaohsiung Medical University, Kaohsiung 80708, Taiwan; 5Division of Pulmonary and Critical Care Medicine, Kaohsiung Medical University Hospital, Kaohsiung 80756, Taiwan; 6Institute of Biomedical Science, National Sun Yat-Sen University, Kaohsiung 80424, Taiwan

**Keywords:** rheumatoid arthritis, chondrocyte, cartilage

## Abstract

Rheumatoid arthritis (RA) is one of the inflammatory joint diseases that display features of articular cartilage destruction. The underlying disturbance results from immune dysregulation that directly and indirectly influence chondrocyte physiology. In the last years, significant evidence inferred from studies in vitro and in the animal model offered a more holistic vision of chondrocytes in RA. Chondrocytes, despite being one of injured cells in RA, also undergo molecular alterations to actively participate in inflammation and matrix destruction in the human rheumatoid joint. This review covers current knowledge about the specific cellular and biochemical mechanisms that account for the chondrocyte signatures of RA and its potential applications for diagnosis and prognosis in RA.

## 1. Introduction

Rheumatoid arthritis (RA) is a chronic inflammatory disease resulting in irreversible joint destruction. It is characterized by synovial fibroblasts (also known as fibroblast-like synoviocytes) activation, inflammation, angiogenesis, and invasion into the adjacent bone and cartilage, resulting in degradation of extracellular matrix (ECM) and bone destruction [1,2,3]. The pathophysiology of RA involves numerous cell types, including macrophages, lymphocytes, chondrocytes and osteoclasts, all of which contribute to the destructive process [4,5,6,7,8]. For many years, other effector cells (lymphocytes, macrophages, synovial fibroblasts, osteoclasts) have been the targets of intensive investigations. In contrast, chondrocytes have received less attention in the past. However, a growing body of evidence suggests that chondrocytes also actively participate in the progressive destructive process of RA. This review would concisely summarize current understanding of the roles played by chondrocytes in RA.

## 2. Chondrocytes in Normal Physiology

Chondrocytes are the only cells in cartilage [9] and are the only cell type that produces and maintains the cartilaginous matrix [10]. Cartilage acts as a key component of synovial joints, consisting of chondrocytes and a dense and highly organized ECM synthesized by these chondrocytes, which contains multiple matrix protein, such as type II collagen and glycosaminoglycans [11].

In addition to ECM, chondrocytes also synthesize lubricin/proteoglycan-4 (PRG4), a glycoprotein that has multifaceted functions including boundary lubrication, which results in reduced friction between apposed cartilage surfaces. Moreover, PRG4 also possesses the capability to suppress inflammatory cytokines which induce proliferation of RA synovial fibroblasts [12,13,14]. In human, loss-of-function mutations in PRG4 result in human autosomal recessive disorder called camptodactyly-arthropathy-coxa vara-pericarditis syndrome (CACP), which is characterized by progressive joint failure associated with noninflammatory synoviocyte hyperplasia and subintimal fibrosis of the synovial capsule [12].

## 3. Chondrocytes in RA

In RA, multiple inflammatory mediators are present in the synovial joint. On the one hand, chondrocytes act as target cells of these inflammatory mediators, resulting in chondrocyte dysfunction. On the other hand, chondrocytes of RA also act as effector cells, exhibiting various alterations that directly or indirectly facilitate joint damage of RA.

### 3.1. Chondrocytes Acting as Target Cells in RA

In RA, multiple proinflammatory molecules are involved, including increased interleukin (IL)-1β, tumor necrosis factor (TNF)-α, IL-6, and IL-17 [15,16,17]. In addition to their well-established actions on immune cells [18], these RA-relevant stimuli result in the molecular activation of catabolic and inflammatory processes in human chondrocytes. For example, multiple cytokines produced by inflammatory cells in RA, including TNF-α and interferon-γ, decrease viability and proliferation of chondrocytes [19]. Enhanced chondrocyte apoptosis is found in the animal model of RA [20] and clinical RA [21]. In addition to facilitating chondrocyte apoptosis, inflammatory mediators also interfere with chondrogenesis. For example, TNF-α inhibits chondrogenic differentiation through p38 mitogen activating protein kinase pathways [22]. Increased CD40 expression on articular chondrocytes of patients with RA is found, and results in enhanced production of cytokines and matrix metalloproteinases from chondrocytes [23].

In conjunction with proinflammatory molecules, stroma cells of synovial joints also actively modulate chondrocytes. In the past, genome-wide microarray analysis of synovial fibroblast-stimulated chondrocytes disclosed a distinct expression profile related to cartilage destruction involving marker genes of inflammation, cartilage degradation, and suppressed matrix synthesis [24]. Synovial fibroblasts and macrophages activated chondrocytes to produce multiple tissue-degrading enzymes (matrix metalloproteinase (MMP)-1, -3, -13 and disintegrin and metalloproteinase with thrombospondin motifs (ADAMTS)-4, -5), and upregulation of inflammatory mediator gene expression (TNF-α, IL-1β, IL-6 and IKBKB) [25]. Synovial fibroblasts also decreased matrix synthesis of chondrocytes [26]. These data all suggest the role of chondrocytes as target cells in RA.

### 3.2. Chondrocytes Acting as Effector Cells in RA

In addition to acting as target cells in RA, evidence also implicated chondrocytes as effector cells in RA directly and indirectly, possibly through releasing multiple enzymes of ECM degradation, facilitating angiogenesis, enhancing inflammation and immune responses, and crosstalk with related cells, as detailed in the following sections.

#### 3.2.1. Chondrocytes Directly Involve in RA Through Releasing Multiple Enzymes of Extracellular Matrix Degradation, Facilitating Angiogenesis, Enhancing Inflammation and Immune Responses

Evidence for this argument comes from production of the collagen and proteoglycan proteinases MMP-1, MMP-3, MMP-10, MMP-12, MMP-13 by chondrocytes [27]. IL-6 stimulates MMP from chondrocytes in addition to enhancing chondrocyte apoptosis [28,29], whereas IL-1 and TNF-α stimulates aggrecanase production [30]. IL-1α and IL-17 stimulate MMP production [31,32]. Likewise, chondrocytes provide factors that activate macrophage-derived pro-gelatinase B (pro-MMP-9) [33].

It has also been proposed that chondrocytes themselves may be a source of pro-inflammatory cytokines, which facilitate the process of joint destruction by increasing the breakdown of tissue and suppressing repair mechanisms. As a result, cartilage is degraded faster than it can be repaired, leading to destruction of the joint [34]. This would lead to impairment of the immune response at the synovium, limitation in the ability of chondrocytes to respond to immune signaling and degrade cartilage, or a combination of both mechanisms. Along with this, fibronectin fragments stimulate expression of multiple cytokines and chemokines by chondrocytes, such as IL-6, IL-8, monocyte chemoattractant protein (MCP)-1, and growth-related oncogene β [35]. Moreover, chondrocytes can also express toll-like receptor (TLR)-1, TLR-2, and TLR-4, and activation of TLR-2 by IL-1, TNF-α, peptidoglycans, lipopolysaccharide, or fibronectin fragments increases the production of MMPs, prostaglandin E (PGE), and vascular endothelial growth factor (VEGF) [34], all of which are mediators in inflammation and angiogenesis, the central step in RA pathogenesis [3]. In the same time, enhanced nitric oxide (NO) production occurs in rheumatoid cartilage [36] and NO is a potent inducer of chondrocyte apoptosis [37] and acts as a proinflammatory and destructive mediator in the process of arthritis [8].

Apart from destruction of ECM, degraded cartilage matrix components are considered as potential autoantigens in the induction and maintenance of RA synovial inflammation [34]. Several cartilage proteins have been demonstrated to act as T-cell autoantigens, stimulate T-cell responses, modulate cytokine secretion in RA [38,39]. In summary, through releasing proinflammatory mediators, angiogenesis inducers, and matrix-degrading enzymes, and promoting immune responses, chondrocytes directly participate in RA pathogenesis.

#### 3.2.2. Chondrocytes Indirectly Involve in RA Through Crosstalk with Related Cells

Earlier report of chondrocyte-synovial fibroblast co-culture showed that the presence of living chondrocytes stimulated synovial fibroblasts to induce cartilage degradation [40]. On top of this, these structural changes in cartilage are important prerequisite for the attachment and invasion of inflamed synovial tissue during destructive inflammatory arthritis [41], suggesting the importance of crosstalk between chondrocytes and synovial fibroblasts.

In the subsequent years, multiple lines of evidence about receptor activator of nuclear factor kappa-B ligand (RANKL), TNF-α and IL-1β, IL-6, IL-8, IL-7, lymphotoxin α, MCP-4, urokinase plasminogen activator (uPA), leukemia inhibitory factor (LIF), serum amyloid A, galectin-3, hypoxia-inducible factor (HIF)-2α expression in chondrocyte suggested potential contribution of these mediators in the crosstalk between chondrocyte and related cells.

a) RANKL: Articular chondrocytes synthesize RANKL and RANKL induces osteoclastogenesis, contributing to juxta-articular bone loss in chronic arthritis such as RA [42].

b) TNF-α and IL-1β: Synovial fibroblasts and macrophages activate chondrocytes to produce TNF-α and IL-1β [25], which stimulate synovial fibroblasts proliferation and invasion [14,43].

c) IL-6, IL-8: TNF also stimulates chondrocytes to release multiple inflammatory cytokines, including IL-6 and IL-8 [44]. IL-6 stimulates RANKL expression by RA synovial fibroblasts [45] and enhances the proliferation of synovial fibroblasts [46], and IL-8 is one of the important contributors to the angiogenic activity of the inflamed RA synovial joint [47].

d) IL-7: Fibronectin fragments stimulate chondrocytes to produce IL-7, and IL-7 stimulates chondrocytes to secrete MMP-13 and release proteoglycan from cartilage explants [48]. In addition, IL-7 drives T-cell-dependent autoimmunity, induces inflammatory cytokines secreted by macrophages/monocytes and leads to tissue destruction [49]. IL-7 also increases responsiveness of CD4+T-cells and lowers the suppressive ability of regulatory T-cells [50], mediating RA pathogenesis by inducing production of potent proangiogenic factors from macrophages and endothelial cells [51]. Furthermore, IL-7 induces bone loss by stimulating osteoclastogenesis that is dependent on RANKL [52].

e) Lymphotoxin α: IL-1β induces lymphotoxin α and enhances adhesiveness of T lymphocytes to chondrocytes [53], and lymphotoxin α stimulates the proliferation of RA synovial fibroblasts, and secretion of cytokines and metalloproteinases from synovial fibroblasts [54].

f) MCP-4: MCP-4 is significantly higher in cartilage from RA patients and enhances the proliferation of synovial fibroblasts by activating the extracellular signal-regulated kinase mitogen-activated protein kinase cascade, thereby leading to joint destruction in RA [55,56].

g) uPA: TNF increases chondrocyte expression of uPA [57], and uPA signaling facilitates synovial fibroblasts invasion into adjacent tissues [58].

h) LIF: IL-1β induces LIF production from chondrocytes [59] and amplifies autocrine loop of IL-6 in synovial fibroblasts [60].

i) Serum amyloid A: Chondrocytes of RA serve as a source of intra-articular acute-phase serum amyloid A protein that induces MMP production and TNF-α expression in synovial tissue [61,62], promotes peripheral blood mononuclear cells recruitment, angiogenesis [63], and synovial cell proliferation [64].

j) Galectin-3: Chondrocytes produce cartilage oligomeric matrix protein, and when synovial fibroblasts adhere to cartilage oligomeric matrix protein, synovial fibroblasts produce increased quantities of galectin-3, which augments synovial inflammation [65,66].

k) HIF-2α: HIF-2α is also upregulated in chondrocytes of RA [67] and when co-cultured with HIF-2α-overexpressing chondrocytes, synovial fibroblasts show increased expression of matrix degradation enzymes (MMP3, MMP9, MMP12, MMP13) and various inflammatory mediators [68] and enhanced migration and invasion, while conditional knockout of HIF-2α in cartilage tissue inhibits pannus formation in adjacent cartilage [69].

Taken together, this evidence highlights the crosstalk between chondrocytes and related cells in the inflammatory condition of RA and show that chondrocytes are not only inflammatory victims but also direct contributor to inflammation and matrix degradation in RA.

## 4. Molecular Mechanisms Underlying Chondrocytes Dysfunction in RA

Cellular dysfunctions including decreased chondrocyte proliferation, enhanced chondrocyte apoptosis, and reduced ECM synthesis in RA have been known for a long time [21,70]. However, the underlying mechanisms and associated molecules related to chondrocyte dysfunction is not completely understood. During the last decades, substantial knowledge has accumulated on the pathogenesis of chondrocyte dysfunction, implicating the involvement of multiple noncoding RNA, signaling pathways, and cellular proteins in RA chondrocytes dysfunction (Figure 1).

a)Noncoding RNA: For example, long noncoding RNA HOTAIR increases chondrocyte proliferation, decreases inflammatory cytokine from chondrocytes, and alleviates RA in the animal model [71], while micro RNA-23a (miR-23a) inhibits IL-17-mediated proinflammatory mediator expression via targeting IκB kinase α (IKKα) in articular chondrocytes [32]. Downregulated miR-26a is found in articular chondrocytes of RA rats, and upregulation of miR-26a reduces swelling and inflammation of joints, diminishes cartilage damage, apoptosis of chondrocytes, and inflammatory injury [72]. Moreover, miR-26a promotes proliferation and counterbalances apoptosis of inflammatory articular chondrocytes [72]. Expression level of miR-27b-3p is decreased in RA, and overexpression of miR-27b-3p significantly reduces the expression of pro-apoptotic protein caspase 3 and increases the expression of anti-apoptotic Bcl-2 in chondrocytes [73].b)Necroptosis pathway: Activation of necroptosis pathway molecules (receptor interacting protein (RIP) 1, RIP3 and mixed lineage kinase domain-like protein phosphorylation (p-MLKL)) are detected in adjuvant arthritis (AA) rat articular cartilage and RIP1 inhibitor necrostatin-1 (Nec-1) could reduce articular cartilage damage and necroinflammation in AA rats [74].c)Pyroptosis pathway: Extracellular acidosis, which accompanies joint inflammation of RA, significantly increases the expression of acid-sensing ion channel 1a (ASIC1a), IL-1β, IL-18, apoptosis-associated speck-like protein (ASC), neuronal apoptosis inhibitor protein, class 2 transcription activator, of the major histocomplex, heterokaryon incompatibility and telomerase-associated protein 1 (NACHT), leucine-rich repeat (LRR) and PYRIN domain (PYD) domains-containing protein 3 (NLRP3) and caspase-1 and mediates chondrocyte pyroptosis [75,76].d)Hedgehog signaling: Expression of hedgehog signal pathway (Shh, Ptch1, Smo, Gli1) in articular cartilage is associated with the severity of cartilage damage in rats with adjuvant-induced arthritis, and hedgehog signal inhibition promotes ECM production [77].e)MAPK pathway: TNF-α activates mitogen-activated kinase (MEK)/ extracellular regulated kinase (ERK) pathway and subsequent early growth response 1 (Egr1) DNA binding activity, which are required for TNF-α regulated catabolic and anabolic gene expression of chondrocytes [78]. Furthermore, acidosis also acts via ASIC1a, leading to intracellular Ca2+ elevation, ERK phosphorylation, culminating in articular chondrocyte apoptosis [79]. MAPK pathway also contributes to IL-1β-stimulated MMP-13 production in RA chondrocytes [80].f)JAK/STAT cascade: IL-6 could enhance acid-induced articular chondrocyte apoptosis, which might partially be involved in regulating the activation of ASIC1a-dependent JAK/STAT pathway [29].g)AP-1 pathway: Stromal cell-derived factor (SDF)-1, significantly higher in RA, acts through CXCR4 to activate ERK and the downstream transcription factors (c-Fos and c-Jun), resulting in the activation of AP-1 on the MMP promoter and contributing to MMP secretion of chondrocytes [81].h)JNK-2 pathway: IL-1 signals via TRAF-6/TAK-1/MKK-4/JNK-2 axis to cause JNK-2-dependent shedding of LRP-1 and subsequent ADAMTS-5-mediated aggrecanolysis [82].Membrane protein: Overexpression of membrane protein aquaporin 4 (AQP4) in articular chondrocytes exacerbates chondrocyte dysfunction of adjuvant-induced arthritis in rats [83].i)Intracellular protein: C/EBPβ mediates expression of MMP-13 in human articular chondrocytes in inflammatory arthritis [84].

In summary, these implicated noncoding RNAs, signaling pathways, and cellular proteins participate in various aspects of disturbed chondrocyte homeostasis, which might provide new therapeutic targets for chondrocyte dysfunction in RA.

## 5. Inhibitors of Chondrocyte Dysfunction in RA

The complexity of molecules involved in chondrocyte dysfunction of RA and advanced knowledge about their roles on chondrocyte present abundant opportunities for therapeutic manipulation in RA. Interestingly, current literature claimed several compounds exhibited the capability to modulate above-mentioned dysregulated pathways in chondrocytes (Figure 2).

Resveratrol, which interfered with lymphotoxin α induced signaling pathways, abrogated inflammatory pathway/degradative/apoptotic changes activated by lymphotoxin α in articular chondrocytes [85], reduced articular damage in the animal model of RA and displayed clinical efficacy [86,87]. Necrostatin-1, which are necroptosis pathway inhibitors, ameliorated articular chondrocyte injury in the animal model [74]. Hyaluronan-inhibited MAPK pathway activation thus suppressed fibronectin fragment-stimulated NO production and reduced IL-1β-stimulated MMP-13 in human RA chondrocytes [80,88,89]. Paclitaxel suppressed AP-1 activity and decreased IL-1-induced MMP-1 and MMP-3 synthesis by chondrocytes [90]. Of these described compounds, resveratrol and hyaluronate showed some clinical benefits in human RA [87,91,92]. Their potential as treatment modality in human RA needs further investigation and validation in larger clinical studies.

Apart from pharmacological treatment of chondrocyte dysfunction in RA, tissue engineering approaches for the repair of joint cartilage have been considered as another alternative choice. In tissue engineering, mesenchymal stromal cells (MSCs) have been of special interest as cell candidates [93]. MSCs, originally isolated from the bone marrow, can also be isolated from various tissues and organs, including cartilage, bone, synovial fluid, synovial membrane, muscle, adipose tissue, amniotic fluid, placenta, and umbilical cord [94]. MSCs play a vital role in tissue repair, and possess high chondrogenic potential [93]. In addition to improving regeneration, MSCs also exhibit various desirable properties such as (a) reduce inflammatory cell infiltration and inflammatory cytokine release; (b) activate regulatory feedback mechanisms [93], and (c) increase chondrocyte proliferation [95]. As such, MSCs are attractive targets for immunomodulation, particularly in the treatment of cartilage injuries and diseases such as RA, since modulation of resident synovial MSCs could lead to control of the inflammatory immune response and restore chondrocyte homeostasis in RA. However, these possibilities in human RA need to be explored by future clinical trials.

To sum up, there is various evidence regarding potential therapeutic targeting of chondrocytes in RA, although some are obtained from in vitro studies and animal models. Whether chondrocyte-directed therapies could be another step toward better treatment of RA needs further study.

## 6. Relationship Between Current Treatment of RA and Chondrocytes

Even though an inflammatory microenvironment in RA resulted in the molecular activation of various pathological processes in human chondrocytes, as mentioned above, these alterations were not irreversible. These inflammatory signatures could be partially reversed by current medication for treatment of RA, such as glucocorticoid, methotrexate, sulfasalazine, leflunomide, hydroxychloroquine, infliximab, etanercept, and tofacitinib [96]. For example, genome-wide expression analysis revealed glucocorticoid and methotrexate normalized expression of catabolic and anabolic mediators stimulated with supernatant of RA synovial fibroblasts in chondrocytes [97]. Hydroxychloroquine, methotrexate and leflunomide restrained IL-1β-induced inducible NO synthase (iNOS) expression and NO production in chondrocytes [98]. Infliximab and etanercept suppressed cytokine-induced expressions of catabolic and inflammatory genes in chondrocytes [99]. Sulfasalazine and tofacitinib neutralized the effects of IL-1β on the protein profiles of chondrocytes [100]. In human articular chondrocytes, the active metabolite of leflunomide raised the production of IL-1 receptor antagonist [101], which ameliorated joint destruction in experimental RA and clinically significantly slowed radiographic progression of RA in human [102]. Overall, although current treatment modality of RA did not target specifically on chondrocytes, they still displayed some favorable effects for RA chondrocytes in the same time of controlling inflammatory responses. In spite of these beneficial characteristics, whether currently approved pharmacological agents for RA can repair cartilage destruction has yet to be demonstrated in longitudinal studies. Therefore, innovative and novel strategies aimed at both reducing inflammation and promoting chondrocyte regeneration are urgently needed to inhibit the progression of RA.

## 7. Utility of Chondrocyte Products as Diagnostic and Prognostic Markers of RA

In the process of inflammation, ECM produced by chondrocytes underwent breakdown and release, and thus was detectable in the peripheral circulation. The amount present in the circulation reflected the extent of cartilage breakdown. Therefore, the presence of these chondrocyte products in peripheral blood potentially served as markers of RA with their quantities proportional to the degree of cartilage destruction in RA. The utility of several proteins as biomarkers have been investigated in previous studies (Figure 3). Serum cartilage oligomeric matrix protein (COMP) has been found to be a strong predictive biomarker for response to abatacept treatment in RA [103]. Furthermore, serum COMP level correlated to disease activity of RA [104] and had superior sensitivity, specificity, and accuracy for diagnosis of RA [105]. Serum collagen type I (TXI) was also associated with RA disease activity [106]. Serum levels of C1M (a product of MMP-cleavage of type I collagen) and C3M (MMP-9-mediated type III collagen degradation product) were able to discriminate between the undifferentiated arthritis and RA diagnosis [107]. Moreover, serum C1M was significantly correlated to disease activity and predicted radiographic progression of RA [108]. Serum C2M, a MMP-generated neo-epitope of type II collagen, allowed discrimination between nonerosive and erosive disease in RA [107]. RA chondrocytes expressed increased YKL-40, which was also called human cartilage glycoprotein-39 (HCgp-39), correlated with disease activity [109,110], predicted radiologic progression [111], and stimulated angiogenesis [112]. Melanoma inhibitory activity (MIA), produced by chondrocytes, was associated with radiographic signs of joint destruction [113]. Temporal course of ratio between type II collagen-related neoepitope (C2C) and type II procollagen carboxy-propeptide (CPII) was also correlated to radiographic progression [114]. In addition, urinary type II collagen (CTX-II) levels predicted long-term radiographic progression in patients with RA [115]. In general, a multitude of chondrocyte products displayed the potential as biomarkers for diagnosis and prognosis stratification in RA.

## 8. Conclusions

Study on the importance of synovial fibroblasts, osteoclasts, and immune cells in RA pathogenesis had substantial progress in the past decade. However, a multitude of evidence supports the notion that chondrocytes are also actively involved in RA pathogenesis [56]. Current evidence suggests that chondrocytes are not just consequences (egg) of RA pathogenesis, but are also causes (chicken) of RA pathogenesis. Furthermore, reports also emphasize the importance of chondrocytes and cartilage in RA. For example, cartilage damage rather than bone erosions appears to be more clearly associated with irreversible physical disability in RA [116], and articular cartilage damage is the most significant determinant of functional impairment in longstanding experimental arthritis [117]. Chondrocyte transplantation reduces inflammation of RA [118]. These altogether highlight the therapeutic potential of chondrocyte manipulation for management of RA. However, current treatment modality of RA mainly targets immune cells rather than chondrocytes. With the advance of knowledge about chondrocyte biology, it is hoped that a drug directing chondrocyte dysfunction could be developed in the future and applied to RA treatment, a disease whereby patients sustain irreversible joint damage despite clinical remission [119].

## Figures and Tables

**Figure 1 ijms-21-01071-f001:**
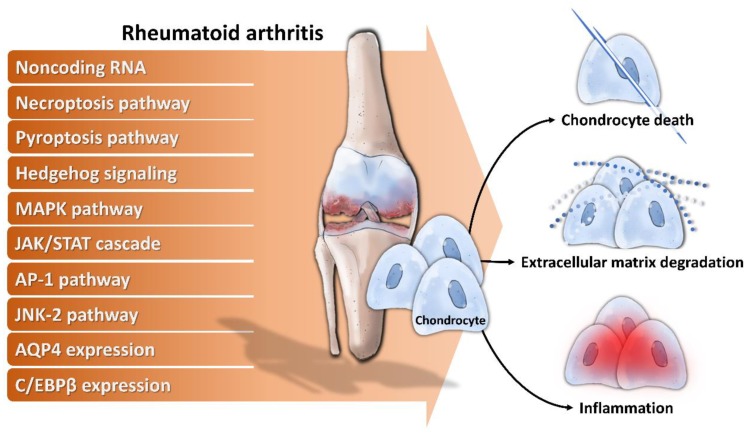
Mechanisms of chondrocytes dysfunction in rheumatoid arthritis. Noncoding RNA and activation of necroptosis pathway, pyroptosis pathway, hedgehog signaling, mitogen-activated protein kinase (MAPK) pathway, Janus kinase/Signal transducer and activator of transcription protein (JAK/STAT) cascade, AP-1 pathway, c-Jun N-terminal kinase 2 (JNK-2) pathway, combined with enhanced aquaporin-4 (AQP4) expression and CCAAT/enhancer binding protein β (C/EBPβ) contribute to increased chondrocyte death, ECM degradation and inflammation.

**Figure 2 ijms-21-01071-f002:**
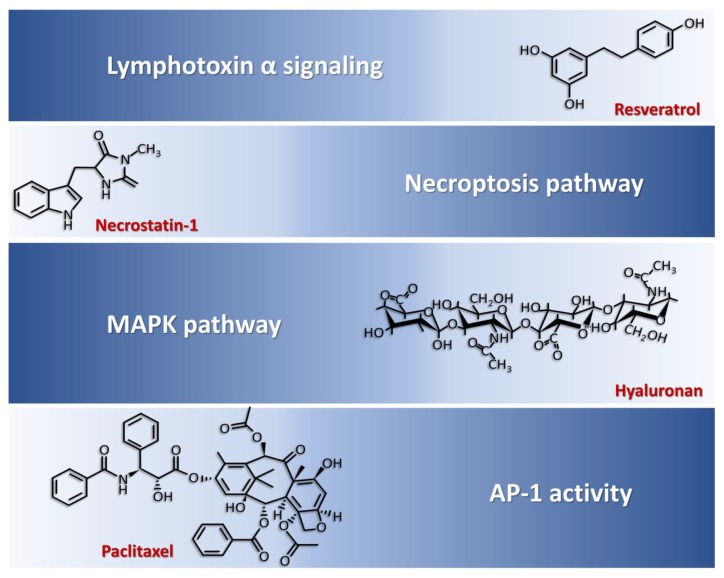
Inhibitors of chondrocytes dysfunction and corresponding targeting pathways. Resveratrol interferes with lymphotoxin α signaling, necrostatin-1 inhibits necroptosis pathway, hyaluronan blocks MAPK pathway, and paclitaxel suppresses AP-1 activity which contribute to chondrocyte dysfunction in rheumatoid arthritis.

**Figure 3 ijms-21-01071-f003:**
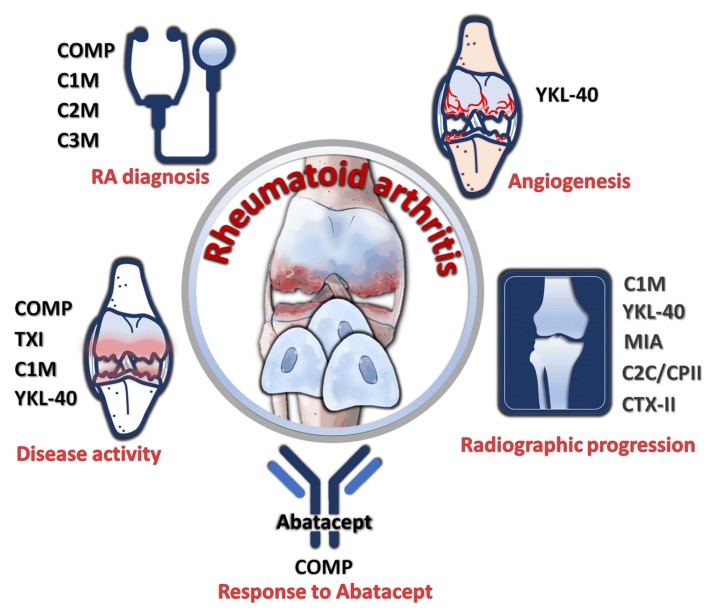
Utility of chondrocyte products as diagnostic and prognostic markers. Cartilage oligomeric matrix protein (COMP), collagen type I (TXI), C1M, C2M, C3M, YKL-40, MIA, C2C/CPII ratio, and type II collagen (CTX-II) were reported to have diagnostic prognostic significance for disease activity, treatment response, radiographic progression, and angiogenesis.

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
