# Peer review of "Dual Role of Chondrocytes in Rheumatoid Arthritis: The Chicken and the Egg"

_ijms, 2020, doi:10.3390/ijms21031071_

Round 1

Reviewer 1 Report

Tseng et al. summarized in this mini-review the recent progress of roles played by chondrocytes in rheumatoid arthritis (RA). They proposed an idea of developing drugs to directly target chondrocyte dysfunction in the future for the RA treatment. The manuscript is well written. I don’t have any further comments but found that in Lines 128 and 129 “RANKL has the capability to enhance osteoclastogenesis” is inaccurate. RANKL induces osteoclastogenesis, but not enhance.

Author Response

I don’t have any further comments but found that in Lines 128 and 129 “RANKL has the capability to enhance osteoclastogenesis” is inaccurate. RANKL induces osteoclastogenesis, but not enhance.

Response: We appreciated your comments. We corrected the statements of Line 128-129 accordingly.

Reviewer 2 Report

Interesting review, well organized and full of information about the subject. However, more figures are required just to visually inform the reader about the mechanisms involved in chondrocyte dysfunction. In particular, sections 5, 6, and 7 need the aid or help with figures ad hoc.

Author Response

However, more figures are required just to visually inform the reader about the mechanisms involved in chondrocyte dysfunction. In particular, sections 5, 6, and 7 need the aid or help with figures ad hoc.

Response: We appreciated your comment. We added Figure 2 to highlight molecules which inhibited contributing signaling pathways and Figure 3 to present the potential utility of chondrocyte in different aspects of RA based on      your suggestions.